# Rheology, Moisture Distribution, and Retrogradation Characteristics of Dough Containing Peony Seed Oil and Quality of Corresponding Steamed Bread

**DOI:** 10.3390/foods14091505

**Published:** 2025-04-25

**Authors:** Ranhuixin Ma, Sihai Han, Jingzheng Song, Zhouya Bai, Chonghui Yue, Peiyan Li, Libo Wang, Denglin Luo

**Affiliations:** 1College of Food and Bioengineering, Henan University of Science and Technology, Luoyang 471023, China; huloo1220@163.com (R.M.); 17538510950@163.com (J.S.); spbaizhouya@163.com (Z.B.); ychad321@163.com (C.Y.); lipeiyan77@163.com (P.L.); lbwang0728@163.com (L.W.); luodenglin@163.com (D.L.); 2Henan Food Raw Material Engineering Technology Research Center, Henan University of Science and Technology, Luoyang 471023, China

**Keywords:** peony seed oil, steamed bread, staling properties, functional bread

## Abstract

In this study, we added peony seed oil (PSO) to wheat dough and made corresponding steamed breads. Through the dynamic rheological tests of the dough, microstructure analyses, bread quality evaluations, crystallization characteristic experiments, and texture characteristic measurements, we revealed the influence mechanisms of the different contents of PSO on the quality characteristics of the wheat dough and Chinese steamed breads. The results showed that adding PSO at 2% (wheat flour weight basis) had a positive effect on the dough’s viscoelasticity, while the G′ and G″ of doughs with higher contents were decreased. When PSO was added in the range from 2.0% to 4.0% (wheat flour weight basis), the scanning electron microscope observation results showed that the reticular structure of dough was denser. The specific volume of the resulting steamed breads increased, the breads were softer, and their chewability was better. The crystallinity of the steamed bread with added PSO was lower, and the hardness of the steamed bread after 24 h of storage was significantly lower than that of the control group, which proved that PSO could delay the staling of steamed breads. This study provides a new idea for the application of PSO as a dietary supplement.

## 1. Introduction

Peony plants are a widely grown plant in China, where more than 1000 varieties have been cultivated [1]. In addition to varieties with bright colors, pleasant aromas, and high ornamental value, peonies have also been cultivated for edible purposes, such as for making tea and pressing oil. Studies have found that peony seed oil (PSO) contains a variety of nutrients, such as polyphenols, tocopherols, and phytosterols [2]. More notably, PSO is rich in α-linolenic acid (ALA), with a content of more than 40% [3], which is an essential fatty acid for humans and must be obtained from food or dietary supplements. In 2011, for its unique nutritional value, peony seed oil was approved as a new food resource (Ministry of Health, 2011, Number 9, China) [4], allowing its industrial production and public consumption; since then, there has been increasing research on the development and utilization of peony seed oil. Wang, X. et al. [5] detected 11 phytosterols in peony seed oil, among which β-sitosterol was the main one. Phytosterols have anti-inflammatory, antioxidant, and anti-atherosclerosis functions [6,7]. The results of DPPH and ferric-reducing antioxidant power assays showed that PSO has significant antioxidant capacity and oxidative stability [8], and it has been reported [9] that PSO can eliminate the aging effect of D-galactose on mice. Su, J. et al. [10] confirmed that peony seed oil can inhibit fat synthesis and up-regulate fatty acid β oxidation; they also found that the serum cholesterol levels of PSO-fed rats were significantly reduced, which played a role in lowering the blood lipid levels in the rats. At present, the research on PSO mainly focuses on its nutritional properties but is not deep enough, resulting in a relatively early stage for the application of PSO in foods.

Chinese steamed bread (also called steamed bun or Mantou), a staple in traditional Chinese cuisine made from fermented wheat dough, occupies an important position in the Chinese diet structure [11]. Compared with Western-style bread, its main difference lies in its production method: steamed bread involves steaming fermented wheat dough, while Western-style bread is made by baking it in an oven. The temperature used for steaming is relatively low and the water content is higher, which can better preserve various endogenous and added nutrients [12]. Furthermore, due to the absence of the Maillard reaction during the production process of steamed breads, as well as the lack of toxic products such as acrylamide and furan, it is considered a healthier staple food and is thus welcomed by an increasing number of countries and regions [13]. With the changes in people’s lifestyles, and the continuous enhancement of nutritional awareness, the standard of satisfaction is increasing day by day, and the traditional steamed breads on the market have been unable to meet the needs of consumers. Additionally, during storage, steamed bread undergoes various physical and chemical changes, leading to moisture loss and increased hardness, which results in a loss of chewiness and elasticity, making the steamed bread prone to hardening and crumbling [14]. This process is known as the staling of steamed bread [15]. The existence of these phenomena, to a certain extent, limits the development of steamed bread production from home-based to modern industrialized food production. Researchers are also actively studying measures to improve dough and product quality to meet the needs of large-scale and efficient steamed bread production, which is now desired due to social and economic changes and the urbanization of China’s population [15].

Most steamed bread dough is made of kneaded wheat flour, water, and yeast powder, which form a typical viscoelastic system. The starch components are usually located in the gluten skeleton and are uniformly combined with gluten protein to form a stable internal structure, which produces viscoelastic behavior and affects the rheological properties of the dough [16]. The rheological properties of the dough are extremely important and affect its mechanical and technological properties, as well as the quality of the product to a large extent. The dough should have higher elasticity, stronger anti-deformation ability, and better air tightness [17]. In a prior study, it was found that when dough was stirred, the addition of oil helped to entrap air inside it. The presence of oil can enhance the adsorption and combination between the particles of various components inside the dough, improve the dough’s ductility and holding capacity [18], and increase its fermentation volume. Meanwhile, lipids combine with starch in dough to form starch–lipid complexes, which can effectively regulate the rearrangement of amylose and amylopectin [19,20], affect the crystalline properties of steamed bread, thereby delaying starch retrogradation and recrystallization [21], and improve the storage properties of steamed bread. Due to the different characteristics of different oils and fats, the gluten network structure that forms in the dough is different, and the resulting dough plasticity and stability are also different. Therefore, we decided to combine PSO and wheat flour in the production process of steamed bread to explore its feasibility and its influence on the physicochemical properties of the dough and bread products.

The purpose of this study was to investigate the effects of different PSO addition amounts on the rheological properties and microstructure of wheat dough, the interaction between PSO and the gluten network in the dough, and the influence on the dough’s viscoelasticity. Observation of the dough microstructure was performed via scanning electron microscopy (SEM). In addition, the effects of the PSO supplementation amount on the specific volume, color, moisture, texture, and staling of steamed bread were investigated. In this way, the mechanism of PSO’s effect on wheat dough and steamed bread was analyzed, as well as the internal relationships between various dough indices and the texture characteristics of corresponding steamed breads. We have also selected rapeseed oil (RO), which is one of the most common edible oils in the Chinese diet, as a control to compare the differences in the impacts of the two on dough and steamed bread. This study provided new insights into the role of peony seed oil in dough formation and steamed bread processing, which could provide ideas not only for more varieties of new steamed bread products but also for the creation of functional food with peony seed oil.

## 2. Materials and Methods

### 2.1. Materials

The wheat flour and rapeseed oil used in this experiment were purchased from Yihai Wheat Co., Ltd. (Zhengzhou, China). The peony seed oil was produced in Shandong province, China. The active dry yeast was purchased from Anqi Yeast Co., Ltd. (Yichang, China). All chemicals were of analytical grade, unless otherwise noted.

### 2.2. Dough Preparation

Wheat flour, yeast, water, and peony seed oil in amounts of 2%, 4%, 6%, 8%, and 10% (relative to the weight of wheat flour) were mixed together using a multifunctional chef machine (PE4680, Guangdong Liran Electric Appliance Industry Co., Ltd., Zhongshan, China). A dough without peony seed oil was used as control group 1 (C1), and wheat dough with 2% of RO (relative to the weight of wheat flour) was used as control group 2 (C2). Part of the dough was relaxed at 4 °C for half an hour, and another 20 g of dough was frozen at −80 °C and then freeze-dried in a vacuum freeze-dryer (Jingfei Technology Co., Ltd., Hangzhou, China) for 24 h so as to be used in subsequent experiments.

### 2.3. Rheological Properties

The rheological properties of the dough were determined using a model DHR-2 rotational rheometer (TA Instruments, New Castle, DE, USA). A 4 g sample of dough was weighed onto a plate with a parallel plate geometry (40 mm diameter) and a 2 mm gap. The exposed sample was coated with a thin layer of silicone oil to prevent it from drying during the test. The linear viscoelastic region of the sample was determined by means of a strain scan test from 0.1% to 10.0%, and the frequency scan was performed with a suitable strain value. The frequency scanning parameters were as follows: spacing, 2 mm; strain value, 0.5%; temperature, 25 °C; holding time, 60 s; frequency scanning range, 0.1~40.0 Hz.

### 2.4. Scanning Electron Microscopy

A scanning electron microscope (TM3000, Hitachi Corp., Mito, Japan) was used to observe the microstructure of each freeze-dried dough sample. A small sample (approximately 40 mm × 30 mm × 10 mm) taken from the center of each freeze-dried frozen dough was treated with gold spray, and the transverse section was observed and imaged at 500× magnification under accelerated voltage in backscattered electron (BSE) mode.

### 2.5. Steamed Bread Preparation

The prepared doughs were fermented in the proofing box at 35 °C and 85% humidity for 40 min. The dough was divided into 70 g portions, each of which was kneaded 60 times to expel air bubbles and fermented again for 20 min. The fermented and molded doughs were steamed for 20 min above boiling water to obtain steamed bread. The products were cooled at an ambient temperature (25 °C) for one hour. Samples were taken from the central area of the steamed breads, frozen in a vacuum freeze-dryer for 6 h, and then vacuum-dried for 24 h. After drying, the dried samples were ground into powder with a mortar and passed through a 100-mesh sieve, then put in the drying dish for use.

### 2.6. Steamed Bread Specific Volume

The height of each steamed bread was measured in three different positions using a vernier caliper, and the average value was recorded. The volume was measured via the rapeseed replacement method according to the Inspection of grain and oils-Steamed buns of wheat flour processing quality evaluation (China National Standard GB/T 35991-2018), and the specific volume was calculated by dividing the volume by the weight.

### 2.7. Steamed Bread Porosity

Images of central bread slices were acquired using an image scanner (Cannon IR-ADV 4225, Tokyo, Japan). The color images were converted to 8-bit grayscale binary images with 300 dpi resolution and analyzed for porosity using Image J-win64 software (National Institutes of Health, Bethesda, MD, USA).

### 2.8. Color Analysis

Samples with a thickness of 20 mm were cut from the center of the steamed breads for testing, and their L*, a*, and b* values were detected using a colorimeter (X-rite color i5, Granville, MI, USA). The L* value indicates the brightness of the sample (0 represents black; 100 represents white), the a* value indicates the red–green degree of the sample (−a represents green; +a represents red), and the b* value represents the yellow–blue degree of the sample (−b represents blue; +b represents yellow). The total color change (ΔE) was calculated according to the following formula [22]:$$\Delta E=\sqrt{{\left(∆L*\right)}^{2}+{\left(∆a*\right)}^{2}+{\left(∆b*\right)}^{2}}$$

### 2.9. Moisture Mobility and Distribution Determination (LF-NMR)

The moisture distribution and fluidity of the steamed breads made from frozen dough were determined using a low-field nuclear magnetic resonance spectrometer (LF-NMR, NMI20-015 V-I, Shanghai Xinmate Electronic Technology Co., Ltd., Shanghai, China). The transverse relaxation time (T_2_) of the steamed breads was scanned using the CPMG (Carr–Purcell–Meiboom–Gill) pulse sequence, and the measurement temperature of the instrument was maintained at 32 ± 0.05 °C. In these experiments, the CPMG sequence parameters were as follows: the main value of the radio signal frequency (SF) was 21 MHz, the sampling frequency (SW) was 200 kHz, the repetition sampling waiting time (TW) was 1500 ms, the number of repetition samplings (NS) was 16, the number of echoes (NECH) was 3000, the echo time (TE) was 0.2 ms, the number of signal sampling points (TD) was 120,010, the radio frequency delay (RFD) was 0.02 ms, the gain RG1 was 10, the gain DRG1 was 3, and the gain PRG was 1. Three signals were collected for each group of steamed bread samples made from the dough.

### 2.10. X-Ray Diffraction (XRD) Analysis

Samples of freeze-dried dough and steamed bread were ground into powder and passed through 80-mesh sieves. The sample powders were analyzed using a D8 ADVANCE diffractometer (Bruker Co., Berlin, Germany) with Cu Ka radiation, a tube pressure of 40 KV, a current of 40 mA, a sample scanning angle (2θ) range of 5~50°, a scanning rate of 2°/min, and a sampling step width of 0.02°. The XRD patterns were analyzed using MDI Jade (version 6.0) software, and the crystalline peaks and amorphous areas were quantified.

### 2.11. Steamed Bread Texture

The textural parameters of the steamed breads were determined using a Texture Analyzer (Instron Universal 5544, Instron Co., Canton, MA, USA). The determination of steamed bread samples stored at ambient temperature for 1 h and 24 h was conducted. The steamed breads were cut into cubes (2 cm × 2 cm × 2 cm), and a P/36 cylindrical probe was used. The speed before the test was 1.0 mm/s, the speed during the test was 2.0 mm/s, the compression ratio was 50.0%, the trigger force was 5.0 g, and the measurement interval was 5.0 s. The steamed bread’s textural properties including hardness, chewiness, springiness, and cohesiveness were measured.

### 2.12. Data Analysis

All sample analyses were performed in triplicate. The data processing results are reported as the means ± standard deviations (mean ± SD). The experimental data were processed using Microsoft Excel, SPSS 26 (IBM Inc., Chicago, IL, USA), and Origin2021 software (Origin Lab Co., Northampton, MA, USA). The statistical differences among the samples were determined via one-way analysis of variance (ANOVA), and statistical significance was defined at *p*-values of <0.05.

## 3. Results and Discussion

### 3.1. Dough Dynamic Rheology Frequency Sweep Analysis

The rheological properties of a dough are represented by its viscoelasticity and fluidity, which are important indicators for evaluating the dough’s processing characteristics [23]. The rheological properties are also a comprehensive characterization of a dough’s kneading resistance. The energy storage modulus (G′) is the energy stored in reversible deformation and is a reflection of the dough’s elasticity; the loss modulus (G″), which is the energy lost due to irreversible deformation, represents the dough’s stickiness. Normally, the protein network is mainly supported by glutenin, which provides elasticity to the dough; glutenin is another important component of the dough network and is related to the dough’s viscosity and ductility [24]. The reactions and interactions among different types of gluten proteins are crucial to the formation of a three-dimensional gluten network when the dough is mixed.

The relationships among the G′, G″, tanδ, and frequency values for doughs with different concentrations of peony seed oil added to them are shown in Figure 1. In the examined frequency range (Figure 1a), all dough samples showed a phenomenon in which the energy storage modulus was greater than the loss modulus (G′ > G″); that is, the elasticity of the dough corresponding to each amount of oil content was greater than the viscosity, showing a solid-like property [25], and the viscoelastic network was relatively stable. As the frequency increased, both the G′ and G″ values of the doughs increased compared with the low-frequency state, which may be due to the fact that the network structure of the dough is not completely elastic, and the recovery of the dough network is a slow process after stress [25]. This solid-like elastic behavior of the dough may be attributed to the dominant repulsive force between the starch particles in the dough samples. Compared with those for the control group (C1 and C2), the G′ and G″ values of the dough supplemented with 2% of peony seed oil both increased, indicating that the elasticity of the dough supplemented with a small amount of PSO was improved; this is because the addition of an appropriate amount of PSO can promote interactions between dough components to stabilize the internal structure and enhance the elasticity of the dough. The experimental results of RO were not as good as those of peony seed oil, which might be related to the components of peony seed oil. When the added amount of PSO was 4%, the G′ and G″ values of the resulting dough at low frequency were slightly higher than those of the blank control group, and the dough’s viscoelasticity was lower than that of the control group when the frequency was increased. With a further increase in PSO concentration, within the tested frequency range, the viscoelastic properties of the experimental dough were consistently lower than those of the control group, which exhibited significantly higher fluidity. It is hypothesized that the incorporation of excessive lipid leads to the formation of loosely packed and fluid-like unsaturated fatty acids within the dough matrix [26], which may encapsulate certain protein and starch granules, thereby impeding gluten network development and resulting in decreased elasticity, reduced viscosity, and enhanced fluidity. As shown in Figure 1b, tanδ is the ratio of G″ to G′, which can be used to characterize the structural strength of a material. The smaller the tanδ value, the more stable the structure of the material [27]. The tanδ values in this experiment were all below 1, which indicates that the doughs exhibited the viscoelastic behavior of a solid-like structure; this is typical of pseudogel systems, where the dough is an elastic solid rather than a viscous liquid. When 2% of PSO was added to the dough, the tanδ value was reduced compared to that for the control group, indicating that the dough network with a small amount of PSO was transformed into a more elastic and rigid structure over the entire frequency range [28].

### 3.2. Dough Microstructure

Scanning electron microscopes (SEMs), with their high resolution and ability to visualize three-dimensional structures, have been widely used to observe the microstructures of doughs [29]. The cross-sectional microstructures of doughs were characterized using SEM with BSE mode, which could provide rich information on the surface structure and composition of dough samples as presented in Figure 2. The doughs in this study contained embedded starch particles of different sizes and shapes, broken starch particles, and even gelatinized starch particles. In Figure 2(a1), the micrograph of the blank control group shows a continuous gluten network and starch particles of different sizes dispersed within the dough network structure. The starch particles are flat and oval and have smooth surfaces, without pinholes, cracks, or breaks. There are also holes of different sizes between the starch particles. This is similar to the microstructures described in previous studies [28,30]. Compared with the control group (Figure 2(a1,a2)), the structural characteristics of the wheat dough with 2% and 4% addition levels of PSO were improved (Figure 2b,c). PSO enhanced the smoothness of gluten gels, resulting in a more uniform distribution of starch granules, the completed structure, and the reduced black holes, indicating a tighter dough structure. When more PSO was added (more than 4%), the oil was evenly dispersed within the internal structure of the dough, forming a gluten film covering the surface of large particle components (Figure 2d–f). At the same time, PSO coated the starch particles, promoting particle aggregation, and with an increase in the PSO addition amount (to 8–10%), the aggregation phenomenon became more obvious (Figure 2e,f), which was similar to the effects of the rapeseed oil addition on wheat dough [31].

There are many holes in a gluten gel structure, which is very fragile and easy to break. A uniform distribution of PSO makes the gluten surface smooth, reduces friction, and reduces the hardness of dough. However, when there is excess fat in a dough, the number of pores is reduced, the oil agglutination is not dispersed, the pores are smaller, and the structure is tight and uneven. It may be that excess PSO dilutes the composition of dough, reduces the molecular weight of polymers such as gluten and starch, destroys the ordered conformation of gluten, reduces the cross-linking entanglement among polymer molecules, and finally weakens the stability and ductility of the dough microstructure. This also explains the reason for the change in the dough rheological properties caused by different PSO addition levels from a microscopic perspective.

### 3.3. Effects of PSO Addition on Steamed Breads’ Specific Volume and Color

#### 3.3.1. Specific Volume and Porosity of Steamed Breads

People’s preferences for steamed bread are similar to those for other breads: most people prefer soft, fluffy breads with a good taste and high volume. As shown in Table 1, compared with that for C1, the specific volumes of steamed breads significantly increased with the addition levels of PSO from 2% to 6%, and decreased with the addition amounts of PSO from 6% to 10%. This phenomenon proved that the addition of PSO promoted the fermentation of the dough and made the steamed bread fluffier, which is consistent with previous experimental results [32]. The breads’ porosity increased significantly only when the addition level of PSO was from 6% to 8%. The addition of oil can reduce the friction between internal particles and has a certain protective effect on the gluten network structure. The resulting dough is fluffy and soft, and its holding capacity is improved. With the addition of a small amount of PSO (2% addition level), the height–diameter ratio of the steamed breads increased (Table 1), and more moderately sized pores with denser distribution were observed (Figure 3B). However, when more PSO was mixed into the system, the height–diameter ratio gradually decreased. When the addition amount was 8% and 10%, the height–diameter ratio of the experimental group was lower than that of the control group, and the cross-sections of the steamed breads exhibited larger pores and a flatter profile. This also explains the phenomenon of the steamed breads having a larger specific volume but a smaller height–diameter ratio, along with the control group’s occasional size imbalance with regard to pores. Overall, the pores were relatively small. The addition of PSO allowed a greater entrapment of air during the process of mixing the raw materials, stabilizing the liquid film structure [33], while the oil itself provided adhesion. An appropriate addition amount improved the cohesion of the gluten network structure, and when the gluten network was relatively firm, the height–diameter ratio of the steamed bread was improved. Although the specific capacity of the steamed bread increased with excessive amounts of PSO additions, the pores in the gluten network increased due to the excessive amount of oil, the dilution of the gluten proteins, and gas production during dough fermentation. However, due to the weak gluten network, the structure was destroyed, causing the steamed bread to easily collapse and the height–diameter ratio to decrease. Compared with PSO and RO at the same addition level, the air holes in the cross-section of the RO steamed bread are uneven in size, and the height-to-diameter ratio is also lower than that of the PSO steamed bread.

#### 3.3.2. Color Difference in Steamed Breads

Color is an important factor for consumers choosing steamed bread. The addition of exogenous substances affects the color of steamed bread, thus affecting consumer acceptance [34]. The color change in a product can be determined based on its brightness L*, red–green degree a*, and yellow–blue degree b*, which are used to calculate color change (ΔE) values [35]. The L* value is the brightness parameter: the larger the L* value, the brighter the sample. Steamed bread samples with the addition of PSO exhibited little color variation, with only slight fluctuations in L*. The steamed breads’ L* value, internal tissue state, pore uniformity, and gas wall thickness are related to a certain extent. With an increase in the amount of PSO added to the steamed breads, their specific volume increased, their internal pores increased, and their pores became more uneven, also affecting their brightness to a certain extent. The smaller the value of ΔE, the closer the two colors are. Generally speaking, when ΔE < 1, it is difficult for the human eye to detect the color difference; when 1 < ΔE < 3, the color difference is relatively slight. PSO is light yellow, while RO has a darker color. Therefore, the value of the ΔE of the steamed bread with the same amount of RO addition is larger, and when the amount of PSO addition is within 6%, the color change in the steamed bread is not obvious, and an excessive addition amount affects the color of the products.

### 3.4. Effect of PSO Addition on Moisture Mobility and Moisture Distribution in Breads

The water in steamed bread exists in three modalities: bound water, which is closely associated with starch and protein; weakly bound water, which is relatively loosely bound to components such as gluten proteins but is not completely free; free water, which can freely move within the bread system. LF-NMR is a valuable tool for assessing the interactions among water, starch, and gluten networks. From measurements of the spin relaxation time (T_2_), the states of water molecules with different mobilities can be identified, facilitating an understanding of the water content and its migration within the food matrix [36]. The three peaks, T_21_, T_22_, and T_23,_ respectively, represent the transverse relaxation times of bound water, weakly bound water, and free water; A_21_, A_22_, and A_23,_ respectively, denote the percentages of peak areas corresponding to tightly bound water, weakly bound water, and free water.

The figure presents the variations In the proton signal amplitude and relaxation time of steamed breads with different levels of peony seed oil added to them. In Figure 4a, the *X*-axis represents the relaxation time related to water activity. An increase in the relaxation time implies a reduction in water-holding capacity and an enhancement in water mobility; the *Y*-axis is the proton signal amplitude, and its peak area represents the contents of different types of water. In Table 2, compared with C1, the T_23_ of the experimental groups and C2 were longer, the A_21_ and A_22_ decreased, and the A_23_ significantly increased, indicating that the water mobility escalated in the steamed bread systems with the addition of oil. The interactions between water molecules and starch mainly generate weakly bound water [37]. Due to the hydrophobic nature of lipids, the addition of low levels of PSO (e.g., 2% and 4%) had no significant effect on the weakly bound water content of the samples. As the amount of oil in the system increases, the oil encapsulating the starch particles might influence the binding of other components in the system with water. It is notable, however, that an increase in free water in the steamed bread and an increase in water mobility rendered the texture of the steamed breads relatively softer [38].

### 3.5. Crystal Structure and Relative Crystallinity

The effects of PSO on the structure and crystallization pattern of dough and bread were further analyzed by means of X-ray diffraction. Figure 5a shows the X-ray diffraction patterns of doughs and steamed breads made by adding different amounts of PSO. Compared with that for the control group, the shapes of the diffraction peaks for the dough samples with different PSO addition amounts were similar. All the test samples had obvious shoulder-characteristic diffraction peaks at scanning angles (2θ) of 17° and 18° and single diffraction peaks at 15° and 23°, showing typical A-type starch diffraction patterns [39]. This may be because the addition of exogenous oil did not change the crystal type of starch in the dough. It can also be seen from the figure that with an increase in the amount of PSO, the intensity of the diffraction peaks for the different experimental groups decreased. It may be that with the addition of oil to the dough, part of the starch was coated with the oil, forming a physical barrier and inhibiting the double spiral rearrangement of amylopectin; this is consistent with the conclusion drawn in a previous study [31]. The XRD patterns of all the steamed breads showed an obvious diffraction peak at 20°, representing the characteristic recrystallization peaks of amylose and amylopectin. Figure 5b shows the X-ray diffraction patterns of steamed breads corresponding to doughs from the experimental group with different PSO supplementation levels. All the steamed bread samples had obvious diffraction peaks at 2θ = 20° and peaks with relatively weak intensity at about 13°. This means that during the process of steaming the dough into steamed bread, the position of the diffraction peaks changed greatly, and the starch crystal structure changed from type A to type V. Thus, the lipids in PSO and amylose starch in the wheat dough underwent spiral interactions to produce V-type starch [40].

Crystallinity represents the crystal integrity of the crystalline region of starch and is related to starch gelatinization characteristics, gelatinization behavior, and digestibility. A grade analysis of crystallinity was used to characterize the degree of staling of stored steamed breads; the relative crystallinity results are shown in Figure 5b. In the dough system, the samples supplemented with PSO showed a significant decrease in relative crystallinity compared to the control group, indicating that the presence of PSO is likely to delay the staling of steamed breads. This may be due to the fact that PSO covers part of the starch and protein, destroys the gluten network structure, and blocks water diffusion and transfer processes and starch protein interactions [22]. The slightly higher crystallinity of C2 compared to the experimental group might be caused by the differences in the compositions of RO and PSO.

### 3.6. Texture Profile Analysis (TPA)

The structural properties of steamed breads are usually determined after cooling. This analysis is defined as a combination of sensory perceptions and mechanical and geometric characteristics, which are important indices for evaluating the quality characteristics of steamed bread [41], and usually include hardness, springiness, cohesiveness, chewiness, etc. Hardness is one of the most important indicators; it can reflect the taste of the product but can also reflect its staling condition. According to people’s preferences for steamed breads, they are more inclined to choose products with low hardness. A lower hardness value means that the steamed bread is softer and fluffier, but excessive softness is not conducive to an improvement in steamed bread’s quality [42]. Chewability refers to the energy required to chew a food into something suitable for swallowing, and for steamed bread, it shows the same trend as the hardness [43]. The texture characteristics of the corresponding steamed breads made with different added amounts of PSO are shown in Figure 6. When stored for 1 h, the hardness of the steamed bread with oil was lower than the steamed bread of C1 (Figure 6a), which was consistent with the effect of chia oil on bread hardness [44]. The hardness of PSO steamed breads is lower than that of C2 and shows a decreasing trend with the increase in the PSO addition from 2% to 6%. The reason for this may be that in this range, the gluten network structure of part of the steamed bread is diluted by the oil. The carbon dioxide generated during dough fermentation enlarges the pores of the network structure, making the structure of the steamed bread softer. When the amount of added oil is further increased, more oil covers the starch particles and proteins, inhibits the water absorption of the starch, and inhibits the hydration of proteins. The gluten network is inhibited, thus giving the dough greater strength and increasing the hardness of the steamed bread [45]. It can also be seen from Figure 6b that the chewability of the steamed bread stored for 1 h is consistent with the change trend in their hardness. The cohesiveness of steamed bread is the force that holds the bread together, which reflects the degree of adhesion inside the bread and its ability to resist external damage; that is, the higher the cohesiveness, the softer the taste of the steamed bread and the less it will stick to one’s teeth. Springiness represents the ability of the bread to recover from deformation, which usually helps to improve the bread’s quality. The elasticity and cohesion of the 2% experimental group were the highest [28], proving that the quality of this steamed bread was the best at this time.

Consistent with the results of a previous study [46], the elasticity and cohesion of the steamed breads decreased after 24 h of storage, while their chewability and hardness greatly increased. The hardness of the control group increased by 374%, while far lower increases were observed for the experimental groups with 2% and 4% added PSO (266% and 243%, respectively). The hardness increase rate of steamed bread in the remaining experimental groups was observed to be significantly lower compared with the control group. This proved that the addition of PSO significantly delayed the staling of the steamed breads [47]. It may be that the lipids in PSO can form complexes with starch, preventing interactions between starch molecules. From this analysis of the structural parameters measured after 24 h storage of the steamed breads, it can be concluded that PSO can delay the short-term staling of such breads.

## 4. Conclusions

This study represents the first attempt to incorporate PSO into dough, investigating how PSO affects dough and steamed bread quality within the concentration range of 2% to 10%, along with the underlying mechanisms governing these effects. Meanwhile, by comparing the dough and steamed bread without oil and with added RO, it was found that the dough with the 2% PSO addition had increased values of G’ and G’’ and a more stable viscoelastic network structure. SEM observed that the addition of this amount had an improvement effect on the microstructure of the dough, which was consistent with the results of the dynamic rheological test. When processed into steamed bread, the specific volume of all the groups with PSO added was larger than that of the steamed bread without oil, making the steamed bread fluffier. The addition of PSO increased the content of free water in the steamed bread, increasing the degree of freedom of the water in the system. XRD crystallinity analysis showed that the addition of PSO reduced the relative crystallinity of the steamed bread, and there is potential to delay the aging of the steamed bread. The results of the texture test indicated that the presence of PSO reduced the hardness and chewiness of the steamed bread, significantly improving the texture properties of the steamed bread. It is worth noting that the overall experimental results of the experimental group with 2% of PSO added were better, with a larger height-to-diameter ratio, the smallest total color difference, and good springiness. Compared with the PSO addition at the same amount, RO showed more obvious negative effects on the dough, and the quality of the steamed bread was not as good as that with PSO. The reason is related to the differences in the components of different oils, and it can be further explored in the future.

In conclusion, these findings suggest that controlled PSO incorporation at appropriate concentrations (notably 2%) represents a viable strategy to develop nutritionally enhanced wheat flour products while maintaining desirable sensory attributes. The limitation of this work included the absence of nutritional property assessments of the PSO-incorporated steamed bread. This study provides a preliminary framework for PSO applications in flour-based foods, and the relevant contents warrant further exploration to advance PSO utilization in functional food research.

## Figures and Tables

**Figure 1 foods-14-01505-f001:**
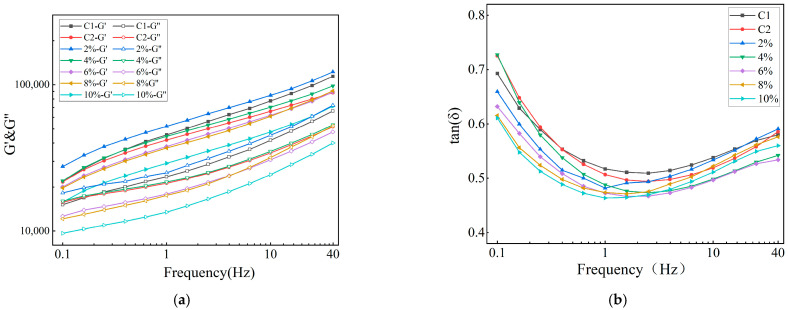
Effects of PSO on the rheological properties of dough: (**a**) elastic modulus G′ and viscous modulus G″; (**b**) loss tangent tanδ.

**Figure 2 foods-14-01505-f002:**
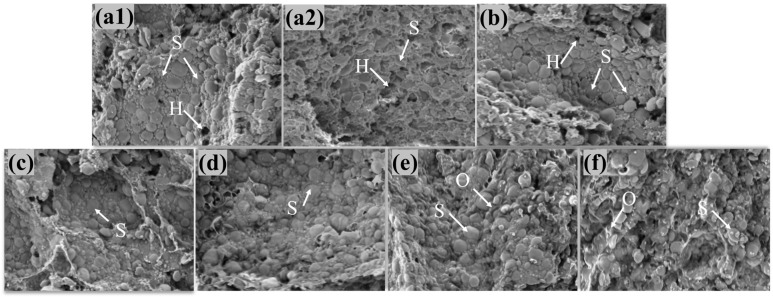
Scanning electron micrographs of dough samples at 500× magnification. (**a1**) and (**a2**) represent the control group (C1 and C2), and (**b**–**f**) represent PSO addition levels of 2%, 4%, 6%, 8%, and 10%, respectively. “S” stands for starch particles, “H” stands for holes of different sizes between the starch particles, and “O” stands for forming a gluten film covering the surface by PSO.

**Figure 3 foods-14-01505-f003:**
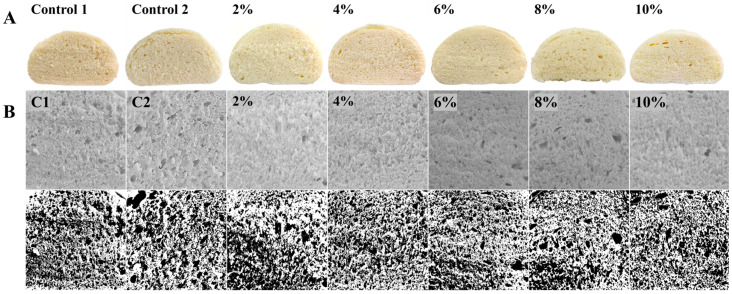
(**A**) The picture represents the cross-section of steamed breads. (**B**) Effect of PSO addition levels on the stoma size and distribution of bread. Note: C1 and C2 are the control group.

**Figure 4 foods-14-01505-f004:**
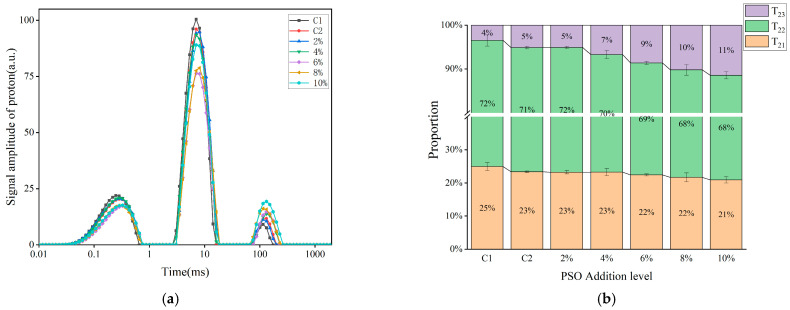
LF-NMR spectra (**a**) and relative percentages (**b**) of different categories of water in steamed breads with different added amounts of PSO.

**Figure 5 foods-14-01505-f005:**
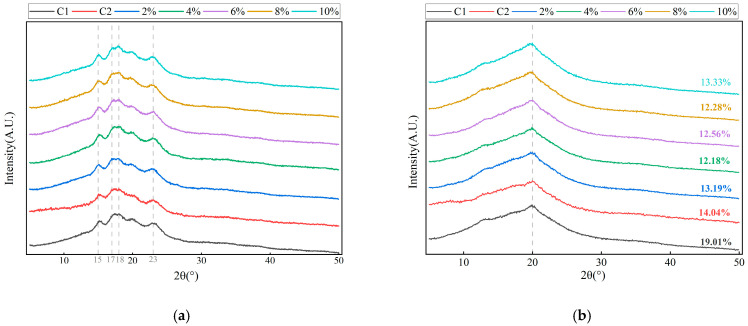
X-ray diffractograms of the control and experimental groups with different levels of PSO addition: (**a**) represents doughs and (**b**) represents steamed breads.

**Figure 6 foods-14-01505-f006:**
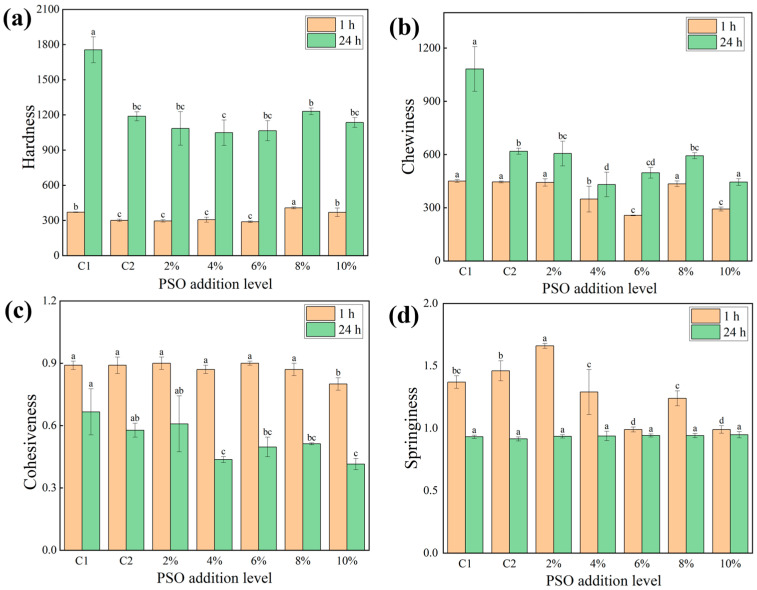
Effects of different PSO addition levels on the textural properties of steamed bread. (**a**–**d**) represent hardness, chewiness, cohesiveness, and springiness, respectively. Different letters in the same type of column diagram indicate significant differences at *p* < 0.05.

**Table 1 foods-14-01505-t001:** Effects of peony seed oil (PSO) on steamed bread quality.

	C1	C2	PSO 2%	PSO 4%	PSO 6%	PSO 8%	PSO 10%
Specific volume (cm^3^/g)	2.49 ± 0.02 ^f^	2.53 ± 0.01 ^e^	2.56 ± 0.02 ^d^	2.61 ± 0.02 ^c^	2.62 ± 0.03 ^b^	2.59 ± 0.02 ^b^	2.52 ± 0.01 ^a^
Aspect ratio	0.55 ± 0.01 ^a^	0.56 ± 0.01 ^a^	0.57 ± 0.00 ^b^	0.55 ± 0.01 ^b^	0.53 ± 0.00 ^b^	0.53 ± 0.00 ^b^	0.51 ± 0.00 ^c^
Porosity (%)	52.94 ± 0.84 ^b^	53.14 ± 0.42 ^b^	52.94 ± 0.53 ^b^	53.40 ± 0.21 ^b^	54.84 ± 0.57 ^a^	54.55 ± 0.30 ^a^	52.93 ± 0.11 ^b^
	Bread color parameters						
L*	77.12 ± 0.45 ^a^	75.85 ± 0.62 ^ab^	77.56 ± 0.59 ^a^	76.99 ± 1.90 ^a^	74.75 ± 0.96 ^bc^	73.74 ± 0.43 ^cd^	72.68 ± 1.37 ^d^
a*	−0.44 ± 0.07 ^a^	−0.59 ± 0.16 ^abc^	−049 ± 0.08 ^a^	−0.55 ± 0.21 ^ab^	−0.74 ± 0.13 ^bcd^	−0.74 ± 0.13 ^cd^	−0.92 ± 0.07 ^d^
b*	12.11 ± 1.18 ^ab^	12.50 ± 0.20 ^a^	11.89 ± 1.04 ^ab^	12.01 ± 0.38 ^ab^	10.53 ± 0.19 ^b^	11.02 ± 1.37 ^b^	10.82 ± 0.28 ^b^
ΔE		1.52 ± 0.46 ^cd^	0.93 ± 0.71 ^d^	1.43 ± 0.88 ^cd^	2.90 ± 0.81 ^bc^	3.75 ± 0.31 ^ab^	4.67 ± 1.25 ^a^

The data are expressed as the mean ± standard deviation (SD) (*n* = 3). Means within a row with different superscript letters are significantly different (*p* < 0.05).

**Table 2 foods-14-01505-t002:** Effects of PSO on lateral relaxation time (T_2_) and peak area (A_2_) of bread moisture.

Technological Parameter	T_21_	T_22_	T_23_	A_21_	A_22_	A_23_
C1	0.26 ± 0.02 ^a^	7.06 ± 0.00 ^b^	114.98 ± 0.00 ^b^	24.93 ± 1.15 ^a^	71.57 ± 1.31 ^a^	3.50 ± 0.19 ^f^
C2	0.27 ± 0.05 ^a^	8.11 ± 0.00 ^a^	120.72 ± 9.94 ^ab^	23.36 ± 0.23 ^b^	71.49 ± 0.29 ^a^	5.15 ± 0.50 ^e^
2%	0.29 ± 0.00 ^a^	7.41 ± 0.60 ^ab^	126.45 ± 9.94 ^ab^	23.23 ± 0.51 ^b^	71.63 ± 0.27 ^a^	5.13 ± 0.24 ^e^
4%	0.27 ± 0.02 ^a^	7.06 ± 0.00 ^b^	126.45 ± 9.94 ^ab^	23.25 ± 1.03 ^b^	70.02 ± 0.92 ^b^	6.73 ± 0.21 ^d^
6%	0.30 ± 0.02 ^a^	7.06 ± 0.00 ^b^	132.19 ± 0.00 ^a^	22.41 ± 029 ^bc^	68.95 ± 0.35 ^bc^	8.64 ± 0.07 ^c^
8%	0.30 ± 0.02 ^a^	7.41 ± 0.60 ^ab^	126.45 ± 9.94 ^ab^	21.68 ± 1.33 ^bc^	68.10 ± 1.20 ^c^	10.22 ± 0.15 ^b^
10%	0.29 ± 0.04 ^a^	7.41 ± 0.60 ^ab^	126.45 ± 9.94 ^ab^	20.93 ± 0.98 ^c^	67.61 ± 0.80 ^c^	11.46 ± 0.18 ^a^

The data are expressed as mean ± standard deviation (SD) (*n* = 3). Means within a row with different superscript letters are significantly different (*p* < 0.05).

## Data Availability

The original contributions presented in the study are included in the article, further inquiries can be directed to the corresponding author.

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
