# Peer review of "Rheology, Moisture Distribution, and Retrogradation Characteristics of Dough Containing Peony Seed Oil and Quality of Corresponding Steamed Bread"

_foods, 2025, doi:10.3390/foods14091505_

Round 1

Reviewer 1 Report

Comments and Suggestions for Authors  

I have been reviewing the manuscript titled “Rheology, Moisture Distribution, and Retrogradation Characteristics of Dough Containing Peony Seed Oil and the Quality of Corresponding Steamed Bread.” In this work, the authors investigate the impact of 2 to 10% peony seed oil content on the quality characteristics of wheat dough and Chinese steamed breads. The experiments were meticulously conducted, and the study was interesting. However, before final acceptance, the authors should address some issues that need correction.

  1. Improving the quality of figures is essential. It is difficult to understand what the figures illustrate.
  2. The benefits of using peony seed oil in bread should be better explained. Additionally, I suggest clarifying the novelty of the work.
  3. Did the authors conduct any sensory and taste analysis? 
  4. There are many editing mistakes that need revision. Throughout the entire manuscript, there is a missing space between the text and the reference number. Additionally, there are extra spaces between words and commas.
  5. Line 2028. “As the added amount of PSO continued to increase, the viscoelasticity of the experimental group was worse than that of the oil-free dough within the range of test frequencies, which is consistent with previous results [23].” What were the results of the mentioned studies compared to the findings obtained?
  6. It lacks a comparison of the study results with other literature findings. I suggest including a table for comparison.
  7. Line 239 is missing a figure number.
  8. The reference list has a different font

Author Response

Comments 1: Improving the quality of figures is essential. It is difficult to understand what the figures illustrate.

Response 1: We agree with your opinion, so we adjusted the pictures in the manuscript.

Comments 2: The benefits of using peony seed oil in bread should be better explained. Additionally, I suggest clarifying the novelty of the work.

Response 2: The benefits of using peony seed oil in bread were explained in “Introduction” (Line 32-49). The novelty of the work was clarified in “Introduction” (Line 101-104).

Comments 3: Did the authors conduct any sensory and taste analysis?

Response 3: The sensory and taste analysis were added. (2.12 Sensory evaluation of steamed bread; 3.7 Sensory evaluation).

Comments 4: There are many editing mistakes that need revision. Throughout the entire manuscript, there is a missing space between the text and the reference number. Additionally, there are extra spaces between words and commas.

Response 4: The editing mistakes throughout the entire manuscript were checked and corrected.

Comments 5: Line 218-220. “As the added amount of PSO continued to increase, the viscoelasticity of the experimental group was worse than that of the oil-free dough within the range of test frequencies, which is consistent with previous results [23].” What were the results of the mentioned studies compared to the findings obtained?

Response 5: According to your questions, I revised this sentence to make our expression more clearly: “With further increase in PSO concentration, within the tested frequency range, the viscoelastic properties of experimental dough were consistently lower than those of the control group, which exhibited significantly higher fluidity. It is hypothesized that the incorporation of excessive lipid leads to the formation of loosely packed and fluid-like unsaturated fatty acids within the dough matrix [26], which may encapsulate certain protein and starch granules, thereby impeding gluten network development and resulting in decreased elasticity, reduced viscosity, and enhanced fluidity”. (line241~247). 

Comments 6: It lacks a comparison of the study results with other literature findings. I suggest including a table for comparison.

Response 6: For the convenience of analysis, we compared the study results with other literature findings at relevant sections. Due to limited time for manuscript revision, we are concerned that including a table for comparison may alter the logical structure of the paper and could not complete the manuscript revision as scheduled by editorial office.

Comments 7: Line 239 is missing a figure number.

Response 7: The missing figure number was added. (line 263).

Comments 8: The reference list has a different font

Response 8: The font of the reference list was adjusted.

Reviewer 2 Report

Comments and Suggestions for Authors

Manuscript ID: foods- 3538026

Title: Rheology, moisture distribution, and retrogradation characteristics of dough containing peony seed oil and quality of corresponding steamed bread

Authors: Ranhuixin Ma, Sihai Han, Jingzheng Song, Zhouya Bai, Libo Wang, Peiyan Li, Chonghui Yue, Denglin Luo

Overview and general recommendation:

The aim of the manuscript was to investigate the effects of different peony seed oil (PSO) addition amounts on the rheological properties and microstructure of wheat dough, the interaction between PSO and the gluten network in the dough, and the influence on the dough’s viscoelasticity.

The topic of using oils from new, unconventional sources is innovative. In my opinion, checking how PSO oil will affect the properties of products to which it is added also fits into the creation of functional food.

However, in my opinion the manuscript should be reconstructed, including the research part, detailed comments are attached below.

Major comments

  1. In my opinion, the studies presented in the manuscript were poorly designed. In the current design, the authors compare dough and bread without added oil (control sample) and with added PSO, so the results they will obtain are largely an answer to the question not how peony oil affects the properties of bread but the addition of oil/fat itself. The control sample should be bread with added oil (of course different, more standard) than PSO.
  2. The authors do not provide the number of experiment repetitions, in my experience if there is no such information it means that the bread was prepared only once. Which as we know is not enough repetitions to perform statistical analysis.
  3. Authors should also delve into the analysis of results after performing statistical analysis. Groups that are given in the form of letters (as in the manuscript submitted for review), if they have the same designation (belong to one homogeneous group) - are the same, you cannot write that they are different. The authors repeatedly point out differences that are not present based on the results of statistical analysis.
  4. The figures included in the manuscript are illegible, it is impossible to read the values mentioned by the authors in the Results and discussion part.

Minor comments

  1. Abstract and in line nr 95– what does corresponding mean in the sentence: “… to wheat dough and made corresponding steamed breads”?
  2. Introduction - Since the authors use steam as a heat treatment for the dough in their research, it would be worthwhile to add information on whether and what changes may occur in peony oil as a result of such treatment. Please also justify the choice of oil type. The first part of the Introduction contains information about the health aspects of peony seed oil, but is there any data in the literature that indicates that such oil will also significantly affect the technological properties of the dough or finished bread?
  3. Line 108: “PE4680, Guangdong Liran Electric Appliance Industry Co., Ltd.).” - Add also city and country
  4. Line 132: “vacuum freeze-dryer” – what was the producer?
  5. Line 134: “then put in the dryer for use” - Why in the dryer, what kind of dryer was it?
  6. Line 132: “replacement method” - Is there any literature for this?
  7. Line 146: “A thickness of 20mm was cut from the middle section” - The thickness itself probably couldn't have been cut, rather a sample with a thickness of...
  8. Line 151: “The total color change (.E) was calculated according to the Please add information on what the individual values and delta E value ranges mean, according to the literature? following formula”
  9. Line 172: “Steamed bread texture” - What distinguishing features were marked?
  10. Figure 1 - The grafs are illegible
  11. Line 139 and many other: When referring to a specific graph or table, the authors do not provide its number in the text.
  12. Line 264: “It may be that excess palm oil dilutes” - There is no information that these are studies by other authors, unless the studies are also by the authors of the manuscript but palm oil was mistakenly included?
  13. Line 277: “for the blank control group” – why blank?
  14. Line 278: “significantly increased and gradually increased with the amount of added PSO.” - This sentence is not entirely true, because after all, up to a certain point this parameter increases and after a certain point it decreases?
  15. Line 281: “Meanwhile, the breads’ porosity also gradually increased”; “and the specific volume of the steamed bread is larger” - This is also not true, only 6% and 8% PSO have statistically significantly higher results?
  16. Line 288: “it is not difficult to observe..” - It is not easy to observe at all, especially since the quality of the drawings is not suitable for such observations.
  17. Line 323: “steamed bread increased slightly…” - No, they don't - they are in exactly the same statistical group, which means they are the same, even though 4% is the same?
  18. Line 350: “Compared with that of the control group, the relaxation time of the experimental groups with added PSO increased” - And again this is not a correct conclusion - for T21 and T22 relaxation time are the same. The only differences in T23 are between 0 and 6%.
  19. Line 357: “water (A22) in the experimental” - Only above 6%?
  20. Table 1.: For delta E, statistics are not important, but an interpretation of what such indications mean must be added. Additionally, the table contains results for parameters a* and b* - and the description of the results does not mention them?
  21. Line 420: “in the table” – or in Figure?
  22. Line 421: “with PSO decreased significantly.” – no statistic calculated, how this can be written?
  23. Figure 6 – illegible, no statistical analysis results included.
  24. Conclusions – should be rebuild, also it is not entirely clear what effect was tested in this experiment - in my opinion the effect of the addition of oil in general, it is not known how the addition of peony oil would stand out from the others, because the control group was bread without added oil.
Comments on the Quality of English Language

I included all my comments in the review text.

Author Response

Major comments

Comments 1: In my opinion, the studies presented in the manuscript were poorly designed. In the current design, the authors compare dough and bread without added oil (control sample) and with added PSO, so the results they will obtain are largely an answer to the question not how peony oil affects the properties of bread but the addition of oil/fat itself. The control sample should be bread with added oil (of course different, more standard) than PSO.

Response 1: We agree with your opinion. To compensate for this deficiency, we compared the effects of other vegetable oils [32], such as peanut oil [26] and rapeseed oil [31] on dough in the results and discussions.

Comments 2: The authors do not provide the number of experiment repetitions, in my experience if there is no such information it means that the bread was prepared only once. Which as we know is not enough repetitions to perform statistical analysis.

Response 2: The number of experiment repetitions was provided. (2.13 Data analysis, Line 206)

Comments 3: Authors should also delve into the analysis of results after performing statistical analysis. Groups that are given in the form of letters (as in the manuscript submitted for review), if they have the same designation (belong to one homogeneous group) - are the same, you cannot write that they are different. The authors repeatedly point out differences that are not present based on the results of statistical analysis.

Response 3: The statistical analysis was performed. The statistical differences among the samples were determined via one-way analysis of variance (ANOVA), and statistical significance was defined at P-values of <0.05.

Comments 4: The figures included in the manuscript are illegible, it is impossible to read the values mentioned by the authors in the Results and discussion part.

Response 4: The figures included in the manuscript were improved.

Minor comments

Comments 1: Abstract and in line nr 95– what does corresponding mean in the sentence: “… to wheat dough and made corresponding steamed breads”?

Response 1: The “corresponding” means that the finished steamed breads was made from the wheat dough with the same proportion of PSO added.

Comments 2: Introduction- Since the authors use steam as a heat treatment for the dough in their research, it would be worthwhile to add information on whether and what changes may occur in peony oil as a result of such treatment. Please also justify the choice of oil type. The first part of the Introduction contains information about the health aspects of peony seed oil, but is there any data in the literature that indicates that such oil will also significantly affect the technological properties of the dough or finished bread?

Response 2: The temperature used for steaming is relatively low and the water content is higher, which can better preserve various endogenous and added nutrients such as peony seed oil. Although peony seed oil has been proved a healthcare oil, there are few reports about its effect on the technological properties of the dough or finished bread so far. Therefore, the purpose of this study was to investigate the effects of different PSO addition amounts on the rheological properties and microstructure of wheat dough, the interaction between PSO and the gluten network in the dough, and the influence on the finished bread.

Comments 3: Line 108: “PE4680, Guangdong Liran Electric Appliance Industry Co., Ltd.).” - Add also city and country

Response 3: The city and country were added. (Line 115)

Comments 4: Line 132: “vacuum freeze-dryer” – what was the producer?

Response 4: The producer was added (Line 118).

Comments 5: Line 134: “then put in the dryer for use” - Why in the dryer, what kind of dryer was it?

Response 5: The “drying dish” was used instead of “the dryer”. (Line143).

Comments 6: Line 132: “replacement method” - Is there any literature for this?

Response 6: The literature “National Standard GB/T 35991-2018, China” for “replacement method” was added. (Line 148)

Comments 7: Line 146:“A thickness of 20mm was cut from the middle section” - The thickness itself probably couldn't have been cut, rather a sample with a thickness of..

Response 7: The sentence was modified as “Samples with a thickness of 20mm were cut from the center of the steamed breads for testing”.(Line 156).

Comments 8: Line 151: “The total color change (ΔE) was calculated according to the Please add information on what the individual values and delta E value ranges mean, according to the literature? following formula”

Response 8: The reference literature was added. (Line 163)

Comments 9: Line 172: “Steamed bread texture” - What distinguishing features were marked?

Response 9: The steamed bread textural properties including hardness, chewiness, springiness, and cohesiveness were measured. (Line 192).

Comments 10: Figure 1 - The grafs are illegible.

Response 10: Figure 1 was improved.

Comments 11: Line 139 and many other: When referring to a specific graph or table, the authors do not provide its number in the text.

Response 11: The specific numbers referring to a specific graph or table were marked in the text. (Line 224, 248)

Comments 12: Line 264: “It may be that excess palm oil dilutes” - There is no information that these are studies by other authors, unless the studies are also by the authors of the manuscript but palm oil was mistakenly included?

Response 12: The writing error “palm oil” was modified as “PSO”. (Line 285)

Comments 13: Line 277: “for the blank control group” – why blank?

Response 13: The “control group” was used instead of “blank control group”. (Line 301)

Comments 14: Line 278: “significantly increased and gradually increased with the amount of added PSO.” - This sentence is not entirely true, because after all, up to a certain point this parameter increases and after a certain point it decreases?

Response 14: This sentence was modified as “the specific volumes of steamed breads significantly increased with the addition level of PSO from 2% to 6%, and decreased with the addition amount of PSO from 6% to 10%”. (line 300-302).

Comments 15: Line 281: “Meanwhile, the breads’ porosity also gradually increased”; “and the specific volume of the steamed bread is larger” - This is also not true, only 6% and 8% PSO have statistically significantly higher results?

Response 15: This sentence was modified as “The breads’ porosity increased significantly only when the addition level of PSO was from 6% to 8%”. (Line 304)

Comments 16: Line 288: “it is not difficult to observe..” - It is not easy to observe at all, especially since the quality of the drawings is not suitable for such observations

Response 16: Figure 3 was improved to observe the moderately sized pores with denser distribution.

Comments 17: Line 323: “steamed bread increased slightly…” - No, they don't - they are in exactly the same statistical group, which means they are the same, even though 4% is the same?

Response 17: This sentence was modified as “Steamed bread samples with PSO addition exhibited little color variation, with only slight fluctuations in L*, and the ΔE showed an increasing trend with increasing PSO addition levels.” (Line 341)

Comments 18: Line 350: “Compared with that of the control group, the relaxation time of the experimental groups with added PSO increased” - And again this is not a correct conclusion - for T21 and T22 relaxation time are the same. The only differences in T23 are between 0 and 6%.

Response 18: This sentence was modified as “Compared with control group, the T23 of the experimental groups were longer, the A21 and A22 decreased, and the A23 significantly increased, indicating that the water mobility escalated in the steamed bread systems with PSO addition”. (Line 366-378).

Comments 19: Line 357: “water (A22) in the experimental” - Only above 6%?

Response 19: This sentence was modified as “Due to the hydrophobic nature of lipids, the addition of low levels of PSO (e.g., 2% and 4%) had no significant effect on the weakly bound water content of the samples”. (Line 370)

Comments 20: Table 1.: For delta E, statistics are not important, but an interpretation of what such indications mean must be added. Additionally, the table contains results for parameters a* and b* - and the description of the results does not mention them?

Response 20: The description of the results mentioned the ΔE, which was calculated by the value of a* and b*.

Comments 21: Line 420: “in the table” – or in Figure?

Response 21: “In Figure 6” was used instead of “in the table”. (Line 429)

Comments 22: Line 421: “with PSO decreased significantly.” – no statistic calculated, how this can be written?

Response 22: This sentence was modified as “When stored for 1h, the hardness of the steamed bread with PSO was lower than the control group (Figure 6a)”.

Comments 23: Figure 6 – illegible, no statistical analysis results included.

Response 23: Figure 6 was improved, and the statistical analysis results was added.

Comments 24: Conclusions – should be rebuild, also it is not entirely clear what effect was tested in this experiment - in my opinion the effect of the addition of oil in general, it is not known how the addition of peony oil would stand out from the others, because the control group was bread without added oil.

Response 24: The conclusions were rebuilt, and the sensory evaluation results were included.

Reviewer 3 Report

Comments and Suggestions for Authors

The manuscript studies the addition of PSO to doughs and bread, demonstrating some limitations in bread. It has been observed that the addition of PSO at different doses modifies its properties. The manuscript's approach is appropriate, as is its structure. The results reflect the methodology.

The authors could make some clarifications, which are detailed below.

  1. In the introduction, could the authors indicate the importance of G′, G′, microscopy and crystallinity in doughs and bread?
  2. “by hand 60 times” The authors could explain this better.
  3. “In the figure …” indicate
  4. In Figure 2, indicate and specify each image
  5. “Compared with those of the control”, specify which one?
  6. The purpose of using “EDX mode” (L126) is not evident in the results or the discussion (section 3.2).

Author Response

Comments 1: In the introduction, could the authors indicate the importance of G′, G", microscopy and crystallinity in doughs and bread?

Response 1: The importance of G′, G", microscopy and crystallinity in doughs and bread were elucidated in the introduction. (1. Introduction: Line 82~86, and Line 92~93)

Comments 2: “by hand 60 times” The authors could explain this better.

Response 2: This sentence was modified as “The dough was divided into 70 g portions, each of which was kneaded 60 rounds to expel air bubbles and fermented again for 20 min”. (Line 136~138).

Comments 3: “In the figure …” indicate

Response 3: “In Figure 2a” was used instead of “In the figure …”. (Line 265)

Comments 4: In Figure 2, indicate and specify each image

Response 4: The “Figure 2a~ Figure 2f” were used to indicate and specify each image. (3.2. Dough microstructure)

Comments 5: “Compared with those of the control”, specify which one?

Response 5: This sentence was modified as “Compared with the control group (no PSO addition)”. (Line 270)

Comments 6: The purpose of using “EDX mode” (L126) is not evident in the results or the discussion (section 3.2).

Response 6: The purpose of using “BSE mode” (“EDX mode” in the original manuscript should be “BSE mode”) was explained. (Section 3.2, Line 261)

Round 2

Reviewer 2 Report

Comments and Suggestions for Authors

Thank you for correcting the manuscript and responding to the reviews.

I am very glad that the entire experiment was performed three times, otherwise the statistics would not have made sense.

However, the authors still have not fully explained whether the obtained results indicate the effect of peony oil or the addition of fat itself.

There is also no interpretation of the delta E results, after all, these differences are calculated to indicate whether the samples differ from the control samples in a visible way.

It is also not entirely clear how they marked "H", "S" and "O" in Figure 2, after all they are also visible in other photos?

Round 3

Reviewer 2 Report

Comments and Suggestions for Authors

I would like to thank the authors for their review. Thanks to the results for rapeseed oil included in the review, it is possible to determine the positive effect of adding peony oil on the structure and quality of dough and bread. However, the lack of this data in the text of the manuscript, in my opinion, reduces its value. However, I leave the decision to the Editor.

Author Response

Comment: I would like to thank the authors for their review. Thanks to the results for rapeseed oil included in the review, it is possible to determine the positive effect of adding peony oil on the structure and quality of dough and bread. However, the lack of this data in the text of the manuscript, in my opinion, reduces its value. However, I leave the decision to the Editor.

Response: We are truly grateful for your taking the time out of your busy schedule to review this manuscript. In accordance with your suggestions, we have added a comparative experiment with RO in the article and updated all the charts and the content of the article. RO serves as the control group in the experiment and is represented by C2. In this revision, the modified content has been marked in red in the manuscript. We sincerely hope that you will take a look at it.